# Plant Density and Location: Optimization of Growth and Quality of Cut Sunflower in Tropical and Subtropical Environments

**DOI:** 10.3390/plants13192810

**Published:** 2024-10-08

**Authors:** Tuane Carlesso Tomasi, Lucas Coutinho Reis, Tiago Ledesma Taira, Jackeline Schultz Soares, Regina Tomiozzo, Lilian Osmari Uhlmann, Nereu Augusto Streck, José Carlos Sorgato

**Affiliations:** 1Faculty of Agricultural Sciences, Universidade Federal da Grande Dourados, Dourados 79804-970, MS, Brazil; tuanetomasi@hotmail.com (T.C.T.); lucasc_reis@hotmail.com (L.C.R.); tiagotaira@ufgd.edu.br (T.L.T.); jackelinesoares@ufgd.edu.br (J.S.S.); 2Department of Crop Science, Universidade Federal de Santa Maria, Santa Maria 97105-900, RS, Brazil; re.tomiozzo@gmail.com (R.T.); uhlmannlilian@gmail.com (L.O.U.); nstreck2@yahoo.com.br (N.A.S.)

**Keywords:** *Helianthus annuus* L., plant population, tropics, subtropics, floriculture

## Abstract

The cultivation of sunflower (*Helianthus annuus* L.) as a cut flower stands out in floriculture due to its aesthetic beauty and commercial value. Understanding how cut sunflower genotypes adapt to different edaphoclimatic regions and management practices is essential to optimize flower quality and productivity. This study aimed to evaluate the effect of plant density and location on the development, growth, and quality of cut sunflower in tropical and subtropical environments. Plant densities of 10, 20, 30, 40, and 50 plants/m^2^ were evaluated in tropical climate and subtropical climate using a randomized block design in a factorial scheme. Results showed significant differences between locations for plant height, capitulum and stem diameter, final number of leaves, leaf area, leaf area index, phyllochron, and the developmental cycle. Plant density significantly influenced these variables except for plant height and developmental cycle. The interaction between location and plant density was significant only for capitulum diameter and final leaf number. The findings indicate that both planting density and location significantly influence the developmental cycle of cut sunflowers, with lower densities favoring more robust plants at harvest. A density of 30 plants/m^2^ is recommended for efficient space use without significantly compromising floral stem quality. All produced stems are marketable, suggesting that adjusting planting density can optimize production without compromising quality, adapting to specific regional conditions.

## 1. Introduction

Ornamental sunflower (*Helianthus annuus* L., Asteraceae) includes genotypes for using as garden plants, cut flower, and potted plant [1]. Its beauty and profitability potential has attracted the interest of producers and entrepreneurs seeking new opportunities for business because of several properties such as short developmental cycle, ease of propagation, plant hardiness, and its attractive inflorescence, widely used in floral arrangements [2,3]. Because of these properties, as a practical application, cut sunflower is one of the species of flowers in the “Flowers for All” project, a Brazilian nationwide extension project that aims to promote floriculture among small and medium producers as a source of income for rural communities, considering the pillars of economic, environmental, and social sustainability [4,5].

The ornamental quality of cut sunflower is influenced by various factors, among which plant density (the number of plants per unit area) and location where it is cultivated. Plant density affects intraspecific competition for resources such as soil nutrients, soil water, and solar radiation so that an inadequate plant density can affect agronomic performance of the crop and flower quality [6,7,8,9]. Location is responsible for providing different environments that affect plant development and growth, which may result in consequences on the quality of cut sunflower. Previous studies have shown the effects on agronomic performance of sunflowers due to the cultivation location, influencing the morphophysiological characteristics of the plant and crop developmental cycle because of different meteorological variables among locations such as precipitation, solar radiation, and above ground and soil temperature [10,11,12]. Therefore, the hypothesis in this study is that there is a plant density × location interaction and that the optimum agronomic plant density may differ among locations in cut sunflower.

Most of the flowers in Brazil are produced in the state of São Paulo [13]. In a large country like Brazil, the concentration of production in a single state or region has advantages and disadvantages. Among the disadvantages is that the chain from producers to consumers is long, which impacts directly on flowers price for consumers. Therefore, it is important to identify other regions of Brazil, which has subtropical, tropical and equatorial climates, with potential to produce high quality flowers and therefore shorten the chain. The objective in this study was to evaluate the effect of plant density and cultivation location on the development, growth, and quality of cut sunflower in a tropical and in a subtropical environment.

## 2. Results

### 2.1. Analysis of Variance

Analysis of variance showed that the cultivation location has a significant influence (*p* < 0.05) on all the analyzed variables (Table 1), which include plant height (PH), capitulum diameter (CD), stem diameter (SD), final leaf number (FLN), leaf area (LA), leaf area index (LAI), phyllochron (PHY) and the developmental cycle from sowing to harvest in days (SO-HA). This suggests that the specific environmental and climatic conditions of each cultivation site play an important role in the development and growth of sunflower plants. On the other hand, planting density showed significant differences for most variables (CD, SD, FLN, LA, LAI, PHY), except for PH and SO-HA, indicating that although planting density influences important aspects of sunflower growth, such as stem diameter and leaf area, it does not significantly affect plant height nor the duration of the developmental cycle. Additionally, there was a significant interaction between the location and planting density for the variables CD and FLN, highlighting that the effect of plant density on these variables is influenced by the location where the crop grows (Table 1).

### 2.2. Location-Specific Results

Analyzing the effect of location, PH, SD, LA, LAI, and PHY had the highest values in Dourados/MS (Table 2). The mean PH observed in Dourados was 151.31 cm, 15% higher compared to the PH in Santa Maria/RS (128.60 cm). The mean SD in Dourados was higher (14.30 mm) compared to Santa Maria (12.55 mm), representing an increase of 12.2%. The mean LA in Dourados was 2594.68 cm^2^/plant, being 22.9% larger than in Santa Maria (2001.11 cm^2^/plant). The LAI in Dourados (7.70 cm^2^ leaf/cm^2^ soil) was 22.6% larger than in Santa Maria (5.96). The mean PHY in Dourados was 22.63 °C day leaf^−1^, while in Santa Maria PHY was 17.72 ºC day leaf^−1^, 21.7% lower.

The highest values of CD, FLN, and SO-HA were observed in Santa Maria. The mean CD in Santa Maria was 60.12 mm, compared to 56.74 mm in Dourados, about 5.6% higher. The mean FLN in Santa Maria was 25.03 leaves, while in Dourados was 21.71 leaves, indicating about 3 leaves less in the tropics. The SO-HA in Santa Maria was 53.40 days, while in Dourados it was about 5 days less (48.70 days) (Table 2).

### 2.3. Plant Density-Specific Results

The highest values of CD, SD, FLN, and LA were observed at the lowest plant density (D10), being 74.18 mm, 18.09 mm, 24.48 leaves, and 4340.47 cm^2^/plant, respectively, decreasing as plant density increased (Figure 1A–D). The LAI and PHY showed higher values at the D50 density, with values of 7.99 cm and 22.60 °C day leaf^−1^, decreasing as plant density increased (Figure 1E,F).

### 2.4. Interaction Effects

The interaction between location and plant density was significant for the variables CD and FLN. The CD showed the highest values in Santa Maria and Dourados at D10, being 80.20 mm and 68.16 mm, respectively (Figure 2A). The FLN in Santa Maria also showed the highest results, with 27.05, while in Dourados, the highest FLN was observed at D50 (22.08) (Figure 2B).

### 2.5. Developmental Phases

The duration (in °C day) of the developmental phases from sowing to R1 (SO-R1), from sowing to R4 (SO-R4) and from sowing to R5 (SO-R5) was higher in Dourados/MS (521 °C day, 848.8 °C day, and 886.65 °C day, respectively) than in Santa Maria/RS (329 °C day, 649.75 °C day and 689.05 °C day, respectively) (Figure 3A), whereas the duration of the developmental phases in days was the opposite (Figure 3B), being 20 days, 37 days and 49 days in Dourados/MS, and 23 days, 41 days and 52 days in Santa Maria/RS, respectively. These results indicate that the cultivation location influences the cycle duration, which should be considered, especially in commercial plantings and when cultivated in specific seasons.

### 2.6. Environmental Conditions

Dourados/MS had higher precipitation, reaching 63.4 mm on the DOY (Day of Year) 332 while in Santa Maria/RS, in the same period, had 9.2 mm. The lowest temperatures were recorded in Santa Maria/RS on DOY 298, with a minimum air temperature of 9.6 °C, while in Dourados/MS the minimum air temperature was 13.3 °C (Figure 4 and Table 3).

The solar radiation, and mean and maximum temperatures were higher in the Dourados/MS with an average of 23.9 MJ/m^2^ day and a maximum temperature of 36.1 °C on DOY 327. Photoperiod (HRLT) was longer in Santa Maria/RS, reaching 14.8 h/day on DOY 341, while in Dourados/MS the longest photoperiod was 14.2 h/day on the same day (Figure 4 and Table 3).

### 2.7. Floral Stem Quality

Regarding the quality of the floral stems, according to the Veiling-Holambra marketing parameters, the variables PL and SD met the quantitative standards for all plant densities and in both locations. However, the standards for CD were observed only in D10 when cultivated in Dourados/MS and in Santa Maria/RS in D10 and D20.

## 3. Discussion

Comparing the agronomic performance of cut sunflower in the two locations, which represent the tropics (Dourados/MS) and subtropics (Santa Maris/RS) of Brazil, that the environmental conditions in Dourados/MS resulted in higher PH, SD, LA, LAI, and PHY. This trend suggests that the tropical environment, characterized by higher temperatures and greater solar radiation, promotes faster growth and more efficient radiation use by plants.

Plant density played a crucial role in the results. In general, plants cultivated at a lower density (D10) showed greater CD, SD, FLN, and LA, while the highest density (D50) favored LAI and PHY. This reflects the complex dynamics of intraspecific competition, where plants at a lower plant density can better use the available resources, leading to greater growth.

The cut sunflower cultivated in Santa Maria/RS stood out in terms of CD and FLN. The higher meteorological variability and cooler temperatures in the Subtropics of Brazil seem to have positively influenced these variables, resulting in a larger capitulum diameter and number of leaves, with a longer developmental cycle.

The location × plant density interaction shown in this study revealed interesting results for CD and FLN. In Dourados/MS, sunflower plants cultivated at D10 showed the highest values of CD, while in Santa Maria/RS, this density and D20 also stood out for FLN. This result indicates that the choice of plant density should be made based on the specific cultivation location, considering the environmental and climatic conditions.

The cultivation location of the cut sunflower hybrid Vincent’s Choice significantly influenced the growth variables, confirming that the crop response is linked to the cultivation environment and plant density. Compared with previous studies, such as those by [14] in Santa Catarina and [15] in Rio de Janeiro, Brazil, with sunflower genotypes for grain and oil where the sunflower plant heigth was 178 and 170 cm, respectively, the values found in our study were lower, highlighting the variability of the culture’s development in different conditions and cultivation locations.

Plant height was one of the variables most influenced by the cultivation location, being greater in plants when cultivated in Dourados/MS. This location had a shorter photoperiod throughout the sunflower developmental cycle, suggesting that favorable conditions, such as temperature accumulated, radiation, and precipitation contributed to accelerated plant growth.

Other growth variables CD, SD, FLN, LA, LAI, are also important for crop development and growth. In this study, these variables were influenced by location, with CD and FLN being greater in Santa Maria/RS, which can be attributed to the greater oscillations in solar radiation and cooler temperatures, demonstrating the influence of these meteorological variables on development and growth variables of sunflower plants.

The SD, along with CD and PH, are variables that indicate the commercial value of cut sunflower. The values found in this study, especially at lower plant densities, align with the quality standards of cut sunflower floral stems of the Veiling Holambra Cooperative, indicating that the produced stems meet the market demands. The phyllochron is an important variable of plant development during the vegetative phase as is a measure of the rate that new leaves appear on the stem, which are essential for the interception and absorption of solar radiation. A higher phyllochron indicates a longer time interval for a new leaf to appear and vice-versa [16]. In this study, it was observed a higher phyllochron in Dourados/MS and, consequently, a longer time interval for each leaf to appear. This explains why the final number of leaves was lower in Dourados than in Santa Maria.

The temperature difference between the two locations, along with plant density, can significantly influence the crop developmental cycle. The increase in plant density can lead to greater shading, directing more photoassimilates towards plant height growth up to a limit, after which resource scarcity can compromise development [17,18]. Air temperature directly influences the phenology of agricultural crops, especially as it is associated with the rate of photosynthesis, likewise affecting the phyllochron, influencing the crop developmental cycle [19]. In Dourados/MS, the higher temperatures and reduced photoperiod during the crop developmental cycle may have contributed to accelerating its development by about 5 days. However, when the duration of the developmental cycle is in thermal time (°C day), the plants cultivated in Dourados/MS presented a longer cycle. This demonstrates the importance of defining which time (calendar days or °C day) is used for defining the duration of developmental phases.

In both locations, the duration of developmental cycle of the cut sunflower was shorter than for sunflower genotypes grown for grain and oil purposes. While our results varied from 49 to 52 days, [2], working with the cultivars BRS Oasis, BRS Refúgio M, and BRS Paixão M in Chapecó—Santa Catarina, reported a developmental cycle varying from 59 to 81 days [20]. noted an inverse relationship between air temperature and developmental cycle duration, that is, higher average air temperatures during the cycle tend to reduce its duration.

Plant density affects intraspecific competition, determining variables of economic yield [7]. Or results indicate that a plant density of 50 plants/m^2^ results in a smaller capitulum diameter and stem, emphasizing the need to consider the growth and production potential of each genotype in different edaphoclimatic conditions to determine the optimum agronomic plant density [21,22]. Ref. [11] also observed that the plant density in sunflower varies according to regional environmental conditions, influencing crucial characteristics for the commercialization of cut sunflower, such as the capitulum diameter. Higher planting density intensifies intraspecific competition, potentially reducing the availability of nutrients for growth [17]. Ref. [23] noted a reduction in the number of capitulum within the standards for the sunflower variety IAC Uruguay with the increase in planting density, attributing this to greater growth in height and the number of normal capitulum, intensifying intraspecific competition.

For the final leaf number, Ref. [24] obtained an average of 23 leaves/plant, a value close to that found in this study in both locations. However, these numbers were lower than the 28 leaves reported by [25] in Campinas, São Paulo.

The results obtained in this study highlight the complex interaction between edaphoclimatic factors and plant density on growth variables in the cultivation of cut sunflower. It has become evident that the cultivation environment plays a crucial role in the development of the crop, with variables such as temperature, radiation, and photoperiod directly affecting the developmental cycle. Moreover, plant density had a considerable impact on commercial characteristics of the flower stem, such as capitulum and stem diameter.

Intraspecific competition at higher plant densities can result in a limited allocation of resources, hindering growth, while lower densities may favor more robust plants at the end of the developmental cycle. These findings are of high practical relevance for cut sunflower producers, offering valuable insights for choosing the optimum plant density and effectively managing edaphoclimatic conditions, aiming at optimizing production. Furthermore, the results emphasize the importance of taking into account regional and environmental variations when planning the production of cut sunflower, adapting cultivation to each location. This strategy can not only enhance the quality of the floral stems but also contribute to the stability of supply throughout the year, positively impacting market prices.

Finally, it is important to emphasize that the focus of this study is on optimizing the management of cut sunflower in tropical and subtropical regions of Brazil, allowing family farmers to cultivate and market ornamental sunflower locally, fostering income generation for these small producers. Therefore, all the stems produced, regardless of plant density and cultivation location, can be marketed without significant losses in the quality of the floral stems. With the data presented in this study, the use of a plant density of 30 plants/m^2^ is recommended, which is pretty close to the plant density of 32 plants/m^2^ used in the “Flowers for All” project [4,5], as it allows a better use of space, increasing the number of flower stems per unit area without significant losses in the quality of the floral stems.

## 4. Materials and Methods

### 4.1. Study Locations

The study was conducted in two locations: in the experimental area of Horticulture at the Faculty of Agricultural Sciences (FCA) of the Federal University of Grande Dourados (UFGD), in Dourados/MS state, in the tropics of Brazil) (22°11′45″ S, 54°55′18″ W, altitude 446 m) in Central-West Brazil, and in a farm in Santa Maria/RS state in the subtropics of Brazil (29°41′29″ S, 53°48′3″ W, altitude 96 m) in Southern Brazil (Figure 5). According to the Köppen classification, the climate of Dourados/MS is Aw, tropical with a rainy season in Summer and a dry season in Winter, and in Santa Maria/RS the climate is Cfa, subtropical humid without dry season and with hot Summer [26].

### 4.2. Meteorological Data Collection

Daily meteorological data of minimum and maximum air temperature, precipitation, and solar radiation during the experimental period at Dourados/MS were from a weather station located 9 km from the experiment and missing data were filled from the weather station of the Brazilian National Weather Service (INMET). In Santa Maria/RS the meteorological data were from a weather station of INMET located about 6 km from the farm. 

### 4.3. Plant Material and Experimental Design

The cut sunflower hybrid Vincent’s Choice was used. The experimental design was a 2 × 5 factorial in randomized blocks with four replications. Factor A was location (Dourados/MS and Santa Maria/RS) and Factor B was plant density (D10 = 10 plants/m^2^, D20 = 20 plants/m^2^, D30 = 30 plants/m^2^, D40 = 40 plants/m^2^, and D50 = 50 plants/m^2^). Each plot (replication) was as 1 m wide × 2 m long (2 m) raised bed (0.25 m height), with 4 rows of plants, in a plant spacing of 0.20 m among rows and 0.40 m within plants in each row (D10), 0.20 m within plants in each row (D20), 0.13 m within plants in each row (D30), 0.10 m within plants in each row (D40), and 0.08 m within plants in each row (D50).

### 4.4. Seedling Preparation and Greenhouse Conditions

In both locations, seeds were sown on 15 October 2021 in plastic trays with 128 cells filled with the same commercial substrate Carolina Soil (pH 5.5; electrical conductivity 0.7 mS/cm, density 130 kg/m^3^; composition: peat, vermiculite, roasted rice hulk and limestone), sowing one seed per cell at 1 cm depth. The trays were kept inside a low-density polyethylene plastic greenhouse during 10 days in Dourados/MS during and 11 days in Santa Maria/RS to protect the seeds and seedlings from rain and birds until transplanting. During the period inside the plastic greenhouse, the emergence of the seedlings was counted daily until complete emergence (final stand).

### 4.5. Soil Preparation and Characteristics

The experimental areas were prepared with plowing and harrowing, followed by raising beds 1 m wide and 0.25 m height with a bed-forming rototiller. Soil samplings were taken for chemical and physical tests (Table 4) at a depth of 0–10 cm, and based on the results of the tests, soil correction was performed. The soil in Dourados is classified as Dystroferric Red Latosol (SiBCS), with a clay texture (60%), while the soil in Santa Maria is a type of transition between typical Aluminic Brown-Ashy Argisol and Arenic Red Dystrophic Argisol.

### 4.6. Transplanting and Fertilization

The transplanting point was defined when the cotyledonary leaves were open at a 180° angle, the first pair of true leaves had a blade length between 1 and 2 cm, and the clod was well-formed before transplanting, 1 L m^−2^ of semi-decomposed chicken manure-based organic fertilizer was broadcast in both experimental areas. After incorporating this organic fertilizer, the beds (1 m × 2 m) for each replication were delimited, followed by chemical fertilization of the soil. In Santa Maria, 50 g m^2^ of NPK 05-20-20 was applied on the day of planting, while in Dourados an equivalent fertilization was used, applying 66.7 g m^−2^ of NPK 10-15-15 on the day of planting. The transplanting was done on 25 October 2021 in Dourados/MS and on 26 October 2021 in Santa Maria/RS, with plants arranged according to the plant density treatments described previously.

### 4.7. Irrigation and Crop Management

Right after transplanting, mulching with straw was added to the surface of the soil on the beds, and three plants in each of the two central lines of each plot were randomly tagged with colored wires. Ten days after transplanting, side dressing fertilization was performed with 25 g m^−2^ of urea and 25 g m^−2^ potassium chloride (KCl). During the experimental period, plants were irrigated as needed with drip tubes for approximately 45 min, with a flow rate of 0.003 L min^−1^. Management practices included manual control of weeds, staking with bamboo stakes and twine at 30, 60, and 90 cm heights, and pest control as needed, using chemical insecticides based on fipronil and biological with Beauveria bassiana fungus, following the manufacturer instructions.

### 4.8. Growth Measurements and Developmental Stages

Starting four days after transplanting, the number of leaves on the tagged plants was counted twice a week until the final leaf (final leaf number). A leaf was counted when it blades length was greater than 2 cm. Plant height and leaf area were measured once a week on two tagged plants per plot up to the beginning of the reproductive phase. The greatest length (L) and the greatest width (W) of each leaf were measured, and the area of each leaf (LA) was calculated using the equation for sunflower by [27] as LA = 0.733 (L × W). The plant leaf area was calculated as the sum of the area of each leaf and then the leaf area index (LAI, cm^2^ leaf/cm^2^ soil) was calculated areas as the plant leaf area divided by the area of soil occupied by each plant [28]. On the tagged plants, the data of the following developmental stages during the reproductive phase (R) were determined according to the PhenoGlad developmental scale for cut sunflower [3]: R1 (visible floral bud), R4 (color of the petals of the outer ligulate flowers of the capitulum is visible), and R5 (harvest point—ligulate flowers at 90° with the capitulum disk) (Figure 6).

### 4.9. Harvest and Quality Evaluation

Daily photoperiod, considering the duration of civil twilight of 6° below the horizon was calculated using the algorithm by [29]. The phyllochron (°C day leaf^−1^) as an indicator of leaf appearance rate was calculated for each plot by the inverse of the angular coefficient of the linear regression between the average number of leaves of the six plants in the plot and the accumulated thermal time (ATt, °C day) [30]. The accumulated thermal time (ATt, °C day) was calculated as ATt = ∑TTd where TTd is the thermal time of the day, calculated as TTd = Tmed − Bt, with Tmed being the daily mean air temperature and Bt is the basal temperature of the crop assumed 7.2 °C [31]. The duration of the developmental phases from sowing to R1 (SO-R1), from sowing to R4 (SO-R4), and from sowing to R5 (SO-R5) was calculated in days and in °C day.

When plants reached the harvest point (R5), the tagged plants in each plot were harvested by cutting them at the soil surface and the stems were cut to standardize floral stems with 0.7 m length (Veiling Holambra standard), leaves were dropped off and floral stems were placed in containers with tap water. The floral stems harvested in Dourados were taken to the Laboratory of in vitro Cultivation of Flowers and Ornamental Plants of UFGD and in Santa Maria to the Crop Science Department of UFSM.

At harvest, the total height of the plant, from the insertion of the stem in the soil to the inflorescence, was the diameter of the stem at the height of the cut and the diameter of the floral capitulum were measured. The floral stems were evaluated for quality, including stem tortuosity, stem diameter, and capitulum diameter. For classification, the quantitative standards of [32] were used, classifying the floral stems in the standard 70: stem length of 0.70 m, minimum stem diameter of 0.80 cm, and the capitulum diameter (open flower) of at least 6.0 cm. Stems that did not meet one of the criteria for minimum diameter or stem length were classified as non-marketable.

### 4.10. Statistical Analysis

The data were subjected to analysis of variance, and where significant differences were found, the Bonferroni *t*-test (*p* ≤ 0.05) was applied for the Factor A (location) and regression analysis for Factor B (plant density) using the SISVAR software (Statistical Analysis Program v.5.6. Federal University of Lavras—MG) [33].

## 5. Conclusions

The cut sunflower hybrid Vincent’s Choice is influenced by plant density and cultivation location. In Dourados/MS, representing the tropics of Brazil, there is a reduction in the developmental cycle. The best developed and best quality plants, in both study locations, were observed at a density of 10 plants/m^2^. However, a density of 30 plants/m^2^ is recommended, as it allows better use of space, increasing the number of flower stems per unit area without significant losses in the quality of the floral stems.

## Figures and Tables

**Figure 1 plants-13-02810-f001:**
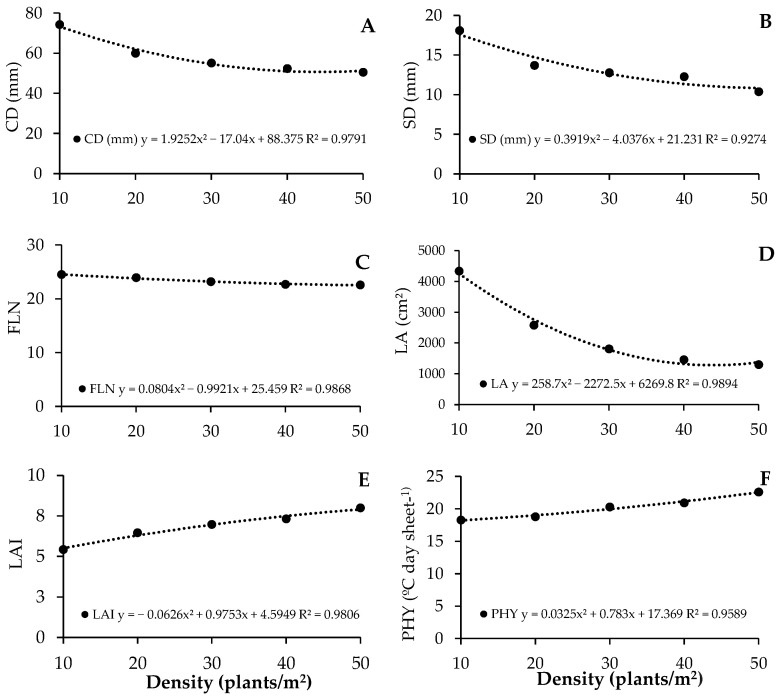
(**A**): Capitulum diameter (CD), (**B**): stem diameter (SD), (**C**): final leaf number (FLN), (**D**): leaf area (LA), (**E**): leaf area index (LAI), (**F**): phyllochron (PHY) as a function of plant density of cut sunflower hybrid Vincent’s Choice.

**Figure 2 plants-13-02810-f002:**
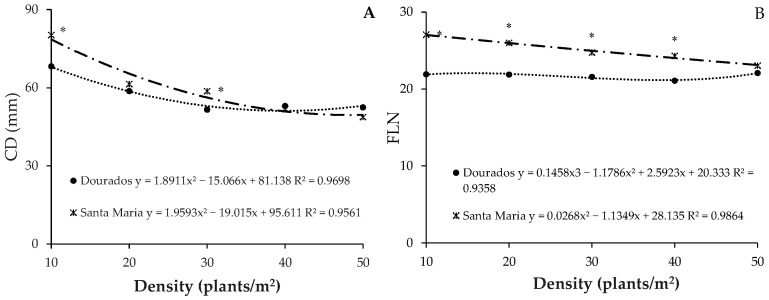
(**A**): Capitulum diameter (CD), (**B**): final leaf number (FLN) of cut sunflower hybrid Vincent’s Choice as a function of plant density in two locations (Dourados/MS and Santa Maria/RS). * indicates statistical difference by Bonferroni test < 0.05 according to the locations within each density.

**Figure 3 plants-13-02810-f003:**
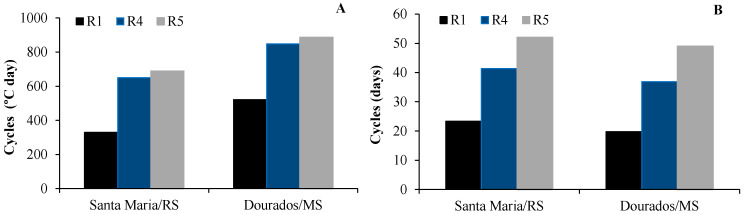
Duration in °C day (**A**) and in days (**B**) of the developmental phases from sowing to R1 (SO-R1), from sowing to R4 (SO-R4) and from sowing to R5 (SO-R5) of cut sunflower hybrid Vincent’s Choice in two locations (Dourados/MS and Santa Maria/RS).

**Figure 4 plants-13-02810-f004:**
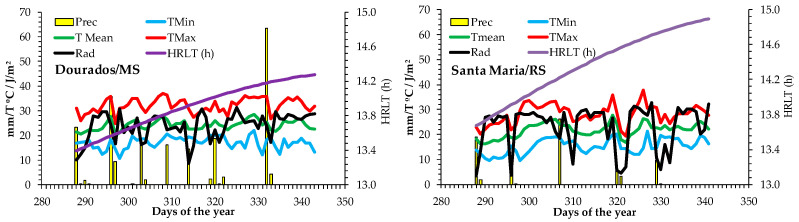
Daily minimum, maximum and mean temperatures (°C), solar radiation (Rad—MJ/m^2^ day), photoperiod HRLT (h), and precipitation (Prec—mm) during the experiment with cut sunflower hybrid Vincent’s Choice in two locations (Dourados/MS and Santa Maria/RS).

**Figure 5 plants-13-02810-f005:**
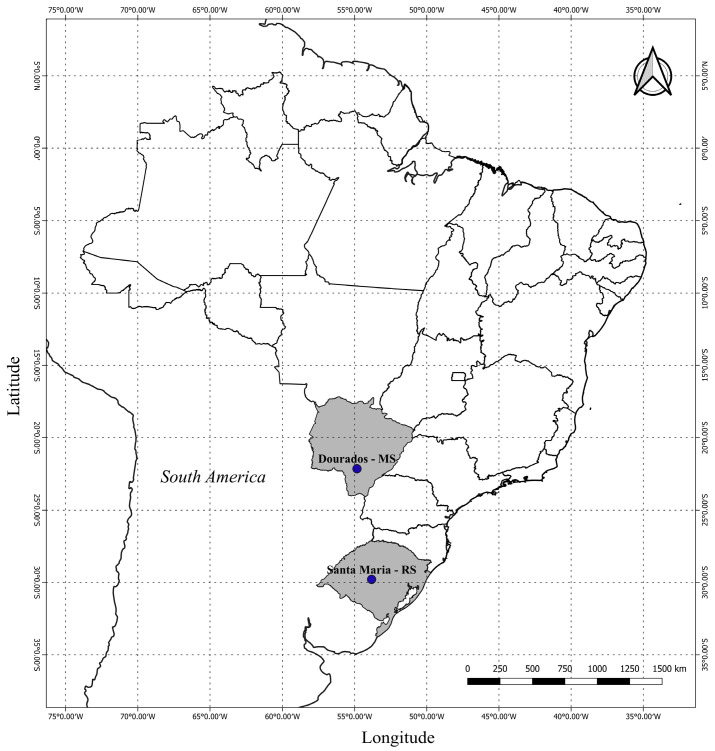
Maps of South America and Brazil, and the States of Rio Grande do Sul (RS) and Mato Grosso do Sul (MS) with the two sites where field experiments with sunflower were conducted (Santa Maria, and Dourados).

**Figure 6 plants-13-02810-f006:**
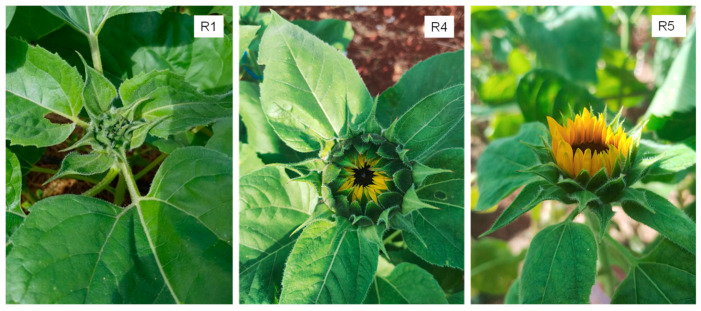
Reproductive stages R1, R4, and R5 in cut sunflower according to the PhenoGlad developmental scale used in the study.

**Table 1 plants-13-02810-t001:** Summary of the analysis of variance for the variables plant height (PH), capitulum diameter (CD), stem diameter (SD), final leaf number (FLN), leaf area (LA), leaf area index (LAI), phyllochron (PHY), and the developmental cycle from sowing to harvest in days (SO-HA) of cut sunflower hybrid Vincent’s Choice in five plant densities (10, 20, 30, 40, and 50 pl/m^2^) and two locations (Dourados/MS and Santa Maria/RS).

VF	DF	Mean Squares							
		PH (cm)	CD (mm)	SD (mm)	FLN	LA (cm^2^/pl)	LAI	PHY	SO-HA (Days)
Location	1	5159.26 *	113.97 *	30.50 *	110.06 *	3,523,271.25 *	30.40 *	241.08 *	220.90 *
Density	4	167.92 ns	721.47 *	65.96 *	5.45 *	12,382,292.63 *	7.44 *	23.97 *	0.16 ns
Location × Density	4	92.67 ns	80.14 *	1.42 ns	7.79 *	181,261.97 ns	0.53 ns	2.05 ns	0.71 ns
Error	30	147.16	18.44	1.15	1.16	130,805.69	0.93	1.70	1.58
Overall Average		139.95	58.43	13.43	23.67	2297.89	6.83	20.17	51.05
CV %		8.67	7.35	7.98	4.61	15.74	14.15	6.46	2.46

*, ns = significant and not significant by the F test at 5% probability, VF = variation factor, DF = degrees of freedom, CV = coefficient of variation, LAI = cm^2^ of leaf/cm^2^ of soil, PHY = °C day leaf^−1^.

**Table 2 plants-13-02810-t002:** Means of plant height (PH), capitulum diameter (CD), stem diameter (SD), final leaf number (FLN), leaf area (LA), leaf area index (LAI), phyllochron (PHY), and cycle from sowing to harvest (SO-HA) of cut sunflower hybrid Vincent’s Choice in two locations (Dourados/MS and Santa Maria/RS).

Variable	Location	
	Dourados/MS	Santa Maria/RS
PH (cm)	151.31 ± 14.56 a	128.60 ± 8.61 b
CD (mm)	56.74 ± 7.02 b	60.12 ± 12.19 a
SD (mm)	14.30 ± 3.02 a	12.55 ± 2.62 b
FLN (leaves)	21.71 ± 0.99 b	25.03 ± 1.73 a
LA (cm^2^/plant)	2594.68 ± 1300.79 a	2001.11 ± 1076.76 b
LAI (cm^2^ leaf/cm^2^ soil)	7.70 ± 1.19 a	5.96 ± 1.31 b
PHY (°C day leaf^−1^)	22.63 ± 2.09 a	17.72 ± 1.96 b
SO-HA (days)	48.70 ± 1.38 b	53.40 ± 0.88 a

Averages followed by the same letters in the line do not statistically differ from each other by the Bonferroni test (*p* < 0.05), ± is one standard deviation of the mean.

**Table 3 plants-13-02810-t003:** Summary of meteorological variables of daily minimum (Min—°C), maximum (Max—°C), and mean temperature, (Mean—°C), daily mean solar radiation, (Rad., MJ m^−2^ dia^−1^), accumulated precipitation, (Prec., mm), and photoperiod (HRLT, h) in Dourados/MS and Santa Maria/RS.

Location	Temperature (°C)			Rad. (MJ m^−2^ dia^−1^)	Prec. (mm)	HRLT (h)
	Min.	Mean	Max			
Dourados/MS	13.3	25.4	36.1	23.9	122.3	14.2
Santa Maria/RS	9.6	21.8	33.6	23.2	76.4	14.8

**Table 4 plants-13-02810-t004:** Soil chemical properties of the sunflower cultivation areas cv. Vincent’s Choice in Dourados/MS and Santa Maria/RS.

Soil Chemical Characteristics		
	Dourados—MS *	Santa Maria—RS **
pH in water	5.97	4.55
pH in CaCl_2_	5.30	--
P (mg/dm^3^)	5.90	88.30
Ca (cmol_c_/dm^3^)	2.72	3.70
Mg (cmol_c_/dm^3^)	1.57	1.70
K (cmol_c_/dm^3^)	0.50	0.52
Sum of bases (cmol_c_/dm^3^)	4.79	5.92
Base saturation (cmol_c_/dm^3^)	55.38	49.80
Organic matter (%)	3.57	2.10

* Methodology Embrapa and IAC; ** Methodology of the Official Network of Soil and Plant Tissue Analysis Laboratories of the States of Rio Grande do Sul and Santa Catarina. Phosphorus (P), Calcium (Ca), Magnesium (Mg) and, Potassium (K).

## Data Availability

The data presented in this study are available in the graphs and tables provided in the manuscript.

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
