# Peer review of "Plant Density and Location: Optimization of Growth and Quality of Cut Sunflower in Tropical and Subtropical Environments"

_plants, 2024, doi:10.3390/plants13192810_

Round 1

Reviewer 1 Report

Comments and Suggestions for Authors

First of all, congratulations on an interesting and carefully worded manuscript. I think that some things could be explained in a little more detail so that the whole text would be even better.   Regarding author citations, I find all cited authors relevant to your research. The references under numbers 3, 4 and 5 are, of course, works previously published by the authors of this manuscript, but they refer to the "Followers for all" project, so I don't see how it would be necessary to move them.   It is necessary to standardize the writing of measurement units. Namely, on page 3 at the bottom of the page (lines 110 and 112) you state cm²/plant, and further on in the text ºC day leaf-1. I believe that the way of writing should be uniform throughout the manuscript (eg page 7, line 234 or page 10, line 354).   Is there a reason why the Vincent's Choice hybrid was chosen for the experiment? Which is why another or another hybrid was not used. It is known that different initial hybrids react differently to their environment, so it might be useful to clarify why exactly that one hybrid was used in this research.   On page 8, in chapter 4.4 Seedling preparation and greenhouse conditions, it is stated that commercial substrate was used. It would be good to state the composition of the substrate.   The research was conducted in two locations: Dourados/MS and Santa Maria/RS. Regardless of the fact that the exact coordinates of both locations are given, it would be good to clarify what the marks MS and RS are.   I encountered several errors such as: 1. page 2, line 57 - ..."vantages and disadvantages." 2. page 5, line 133 - ..."than e in Santa Maria/RS..." 3. page 5, line 134 - ..."whereas de duration"... so please correct it.          

Author Response

Comments 1: Regarding author citations, I find all cited authors relevant to your research. The references under numbers 3, 4 and 5 are, of course, works previously published by the authors of this manuscript, but they refer to the "Followers for all" project, so I don't see how it would be necessary to move them.   

Response 1: We made adjustments in the writting to show the practical application of cut sunflower in the "Flowers for All" project as an ongoing important use of this flower crop in Brazil. Therefore, we would like to keep references 3,4, and 5 as part of the text.

Comments 2: It is necessary to standardize the writing of measurement units. Namely, on page 3 at the bottom of the page (lines 110 and 112) you state cm²/plant, and further on in the text ºC day leaf-1. I believe that the way of writing should be uniform throughout the manuscript (eg page 7, line 234 or page 10, line 354).  

Response 2: Thank you very much for pointing this out. The manuscript has been revised, and the units were standardized according to the international literature in this field. Below some articles using the same units used in the present study:

  • 2134/agronj2013.0187
  • https://doi.org/10.1016/j.sajb.2023.08.006
  • https://doi.org/10.1186/s13007-023-01029-7
  • https://doi.org/10.2135/cropsci1971.0011183X001100050009x

Comments 3: Is there a reason why the Vincent's Choice hybrid was chosen for the experiment? Which is why another or another hybrid was not used. It is known that different initial hybrids react differently to their environment, so it might be useful to clarify why exactly that one hybrid was used in this research. 

Response 3: Thank you very much for the comment. The hybrid Vincent's Choice was chosen because its seeds are readily available in the market for researchers and producers of cut flowers, it does not produce pollen, which increases its durability as a cut flower, it is an early variety that allows for a quick economic return for the producer, and it adapts well to open-field cultivation. In previous experiments conducted at different locations and times of the year in the “Flowers for All” project, this hybrid stood out for its hardiness and good adaptation for year-round cultivation, which is why we decided to use it for the experiment.

Comments 4:  On page 8, in chapter 4.4 Seedling preparation and greenhouse conditions, it is stated that commercial substrate was used. It would be good to state the composition of the substrate.   

Response 4: We agree with this comment. The substrate composition has been added to the text on the page 8, line 305 - 307.

Comments 5: The research was conducted in two locations: Dourados/MS and Santa Maria/RS. Regardless of the fact that the exact coordinates of both locations are given, it would be good to clarify what the marks MS and RS are.  

Response 5: We agree with this comment. We clarify in the text that MS and RS are the states and, to facilitate the identification of the location of the states, Figure 5 has been added to the text to clarify the locations of both sites in South America.

Comments 6:  I encountered several errors such as: 1. page 2, line 57 - ..."vantages and disadvantages." 2. page 5, line 133 - ..."than e in Santa Maria/RS..." 3. page 5, line 134 - ..."whereas de duration"... so please correct it.          

Response 6: Thank you very much for pointing this out. We agree with this comment and correct it.

Reviewer 2 Report

Comments and Suggestions for Authors

A very interesting and relevant study for economic autonomy for farmers/smallholders.

I have very few comments that would make an improvement. I enjoyed reading it!

Line: 57 typo

Line 133 and 134 typos

Figure 3: why is the 2nd x axis HRLT (h) not linear? – i.e. 13, 13, 14, 14 ,15,15

Figure 3 shows that most climatic parameters are pretty similar, apart from the high precipitation episode in Dourados and higher HRLT in Santa Maria. Actual quantities are not reported for both locations, a small table would help to differentiate these

How typical are these differences over several years, given they are described as tropic and subtropic. I understand the limitations of having only one year to study, but should be considered in economic terms of where to cultivate. Perhaps larger graphs would show any other differences/nuances better?

Line 159: should read Fig 3

Line 364: typo @

Soil properties show similar characteristics apart from a very large difference in P content (which I assume is phosphorus? Chemicals should be in full) Is there any comment on this and that it had no effect? Was soil nitrogen not considered?

I have ticked yes to the relevance of references as it is a required field, but I am not qualified to answer this as I do not read Portuguese

Author Response

Comments 1: Line: 57 typo

Response 1: Thank you very much for pointing this out. We correct it.

Comments 2: Line 133 and 134 typos

Response 2: Thank you very much for pointing this out. We correct it.

Comments 3: Figure 4: why is the 2nd x axis HRLT (h) not linear? – i.e. 13, 13, 14, 14 ,15,15

Response 3: Sorry because of this mistake. The Second Y axis of Figure 4 was corrected to obey reviewer answer. The correct numbers are: 13, 13.4, 13.8, 14.2, 14.6 and 15.

Comments 4: Figure 3 shows that most climatic parameters are pretty similar, apart from the high precipitation episode in Dourados and higher HRLT in Santa Maria. Actual quantities are not reported for both locations, a small table would help to differentiate these

Response 4: Thank you for pointing this out. We agree with this comment and, to facilitate the visualization of the climatic parameters, Table 3 has been added to the text.

Comments 5: How typical are these differences over several years, given they are described as tropic and subtropic. I understand the limitations of having only one year to study, but should be considered in economic terms of where to cultivate. Perhaps larger graphs would show any other differences/nuances better?

Response 5: The difference between the crop in the tropics and the subtropics may vary throughout the year and over several years. During the year, the length of the developmental cycle is longer in the subtropics during Winter months and similar to the tropics during Summer months. Over the years, interannual climate variability such that caused by El Nino Southern Oscilation (ENSO) can increase the differences between tropics and subtropics. In order to deal with these differences, farmers are advised to adjust some management practices such as the optimum sowing date in order to have a constant flower production.

Comments 6: Line 159: should read Fig 3

Response 6: Thank you for pointing this out. We correct it.

Comments 7: Line 159: should read Fig 3

Response 7: Thank you for pointing this out. We correct it.

Comments 8: Line 364: typo @

Response 8: Thank you for pointing this out. We correct it.

Comments 9: Soil properties show similar characteristics apart from a very large difference in P content (which I assume is phosphorus? Chemicals should be in full) Is there any comment on this and that it had no effect? Was soil nitrogen not considered?

Response 9: The large difference in phosphorus between both locations is due to the previous use of the experimental area. In Santa Maria/RS, the experiment was conducted in field used for long-term vegetable production with high amount of phosphorus input during its previous use, which was not the case of Dourados/MS location, which was previously used with native grasses with less input of Phosphorus. This previous management explains the large variation in P content between both areas. The full name of chemicals was added at the bottom of Table 4 to obey reviewer comment.  With regards the nitrogen fertilizers, soil organic matter was used to measure the dose of soil cover fertilizer (25 g m-2 of urea and 25 g m-2 potassium chloride (KCl)) as recommend for sunflower in southern Brazil.